# Mid-Pliocene West African Monsoon Rainfall as simulated in the PlioMIP2 ensemble

Ellen Berntell[1], Qiong Zhang[1], Qiang Li[1], Alan M. Haywood[2], Julia C. Tindall[2], Stephen J. Hunter[2], Zhongshi Zhang[3,4], Xiangyu Li[3], Chuncheng Guo[4], Kerim H. Nisancioglu[5,6], Christian Stepanek[7], Gerrit Lohmann[7,8], Linda E. Sohl[9,10], Mark A. Chandler[9,10], Ning Tan[11,12], Camille Contoux[12], Gilles Ramstein[12], Michiel L. J. Baatsen[13], Anna S. von der Heydt[13,14], Deepak Chandan[15], William Richard Peltier[15], Ayako Abe-Ouchi[16], Wing-Le Chan[16], Youichi Kamae[17], Charles J. R. Williams[18,19], Daniel J. Lunt[18], Ran Feng[20], Bette L. Otto-Bliesner[21], Esther C. Brady[21]

[1]Department of Physical Geography and Bolin Centre for Climate Research, Stockholm University, Stockholm, Sweden
[2]School of Earth and Environment, University of Leeds, Woodhouse Lane, Leeds, West Yorkshire, UK
[3]Department of Atmospheric Science, School of Environmental Studies, China University of Geosciences, Wuhan, China
[4]NORCE Norwegian Research Centre, Bjerknes Centre for Climate Research, Bergen, Norway
[5]Department of Earth Science, University of Bergen and Bjerknes Centre for Climate Research, Bergen, Norway
[6]Centre for Earth Evolution and Dynamics, University of Oslo, Oslo, Norway
[7]Alfred Wegener Institute - Helmholtz-Zentrum für Polar und Meeresforschung, Bremerhaven, Germany
[8]Institute for Environmental Physics, University of Bremen, Bremen, Germany
[9]Center for Climate Systems Research, Columbia University, New York, USA
[10]NASA Goddard Institute for Space Studies, New York, USA
[11]Key Laboratory of Cenozoic Geology and Environment, Institute of Geology and Geophysics, Chinese Academy of Sciences, Beijing, China
[12]Laboratoire des Sciences du Climat et de l'Environnement, LSCE/IPSL, CEA-CNRS-UVSQ, Université Paris-Saclay, Gif-sur-Yvette, France
[13]Centre for Complex Systems Science, Utrecht University, Utrecht, The Netherlands
[14]Institute for Marine and Atmospheric research Utrecht (IMAU), Department of Physics, Utrecht University, Utrecht, The Netherlands.
[15]Department of Physics, University of Toronto, Toronto, Ontario, Canada
[16]Centre for Earth Surface System Dynamics (CESD), Atmosphere and Ocean Research Institute (AORI), University of Tokyo, Tokyo, Japan
[17]Faculty of Life and Environmental Sciences, University of Tsukuba, Tsukuba, Japan
[18]School of Geographical Sciences, University of Bristol, Bristol, UK
[19]NCAS, Department of Meteorology, University of Reading, Reading, UK
[20]Department of Geosciences, University of Connecticut, Storrs, CT, USA
[21]Climate and Global Dynamics Laboratory, National Center for Atmospheric Research, Boulder, CO, USA

*Correspondence to*: Ellen Berntell (ellen.berntell@natgeo.su.se)

**Abstract.** The mid-Pliocene Warm Period (mPWP; ~3.2 million years ago) is seen as the most recent time period characterized by a warm climate state, with similar to modern geography and ~400 ppmv atmospheric $CO_2$ concentration, and is therefore often considered an interesting analogue for near-future climate projections. Paleoenvironmental reconstructions indicate higher surface temperatures, decreasing tropical deserts, and a more humid climate in West Africa characterized by a strengthened West African Monsoon (WAM). Using model results from the second phase of the Pliocene Modelling Intercomparison Project (PlioMIP2) ensemble we analyze changes of the WAM rainfall during the mPWP, by comparing with the control simulations for the pre-industrial period. The ensemble shows a robust increase of the summer rainfall over West Africa and the Sahara region with an average increase of 2.5 mm/day, contrasted by a rainfall decrease over the equatorial Atlantic. An anomalous warming of the Sahara Desert and deepening of the Saharan Heat Low, seen in >90% of the models, leads to a strengthening of the WAM and an increased monsoonal flow into the continent. A similar warming of the Sahara Desert is seen in future projections using both phase 3 and 5 of the Coupled Model Intercomparison Project (CMIP3 and CMIP5). Though previous studies of future projections indicate a west/east drying/wetting contrast over Sahel, PlioMIP2 simulations indicate a uniform rainfall increase in that region in warm climates characterized by increasing greenhouse gas forcing. We note that this effect will further depend on the long-term response of the vegetation to the $CO_2$ forcing.

1. Introduction

The mid-Pliocene Warm Period (mPWP; 3.264-3.025 Ma; also known as the mid-Piacenzian Warm Period) is considered to be the most recent past warm climate state, with average global temperatures several degrees above pre-industrial (PI) levels (1.4 - 4.7 °C; Haywood et al., 2020) and atmospheric $CO_2$ concentrations of ~400 ppmv (Badger et al., 2013; Bartoli et al., 2011; Dowsett et al., 2010; Haywood et al., 2020, 2013; de la Vega et al., 2020; Martínez-Botí et al., 2015; Pagani et al., 2010; Raymo et al., 1996; Salzmann et al., 2013; Seki et al., 2010; Tripati et al., 2009; Zhang et al., 2013). Paleoenvironmental reconstructions indicate a warm and humid climate during the mPWP, with elevated sea surface temperatures (SSTs) and surface air temperatures (SATs) especially at high latitudes (Dowsett et al., 2010; Salzmann et al., 2013), forests and grassland expanding into areas at more recent times covered by tundra, and savanna and woodland expanding at the expense of deserts (Salzmann et al., 2008). While much of the research on the mPWP climate focused on global large-scale patterns and the high latitudes (Haywood et al., 2013, 2020; De Nooijer et al., 2020), several studies have emphasized the implications of the warm climate state for tropical climate, showing e.g. an enhancement of the East Asian Summer Monsoon (Wan et al., 2010) and a drying of the Southern Hemisphere tropics and subtropics (Pontes et al., 2020). Analysis off e.g. dust records of the coast of West Africa also indicates a strengthened West African Monsoon (WAM) during the mPWP as well as wetter conditions over West Africa and the Sahara region (Kuechler et al., 2018; Salzmann et al., 2008).

With a paleogeography and atmospheric $CO_2$ concentrations similar to today (Dowsett et al., 2010), the mPWP has long been considered an interesting analogue for near-future climate projections (Chandler et al., 1994; Jiang et al., 2005) and been the focus of many modelling studies (e.g. Haywood and Valdes, 2004; Salzmann et al., 2008). To increase our understanding of the dynamical drivers of the warm climate state, several model simulations have been performed as part of the Pliocene Modelling Intercomparison Project (PlioMIP; Haywood et al., 2010, 2011). Model-data comparisons between the PlioMIP1 (first phase of PlioMIP) simulations and PRISM3 (PRISM – Pliocene Research Interpretation and Synoptic Mapping) reconstructions (Dowsett et al., 2010, 2012, 2013) have shown an underestimation of the high-latitude warming in the mPWP and an overestimation of the warming in the Tropics (Haywood et al., 2013; Salzmann et al., 2013), which has influenced the representation of the WAM within the models (Zhang et al., 2016). PlioMIP1 was later followed up by a second phase (PlioMIP2), representing a more narrow geological time-window (marine isotope stage KM5c, 3.205 Mya) to e.g. facilitate data-model

comparison (Haywood et al., 2016), and though some areas of concern still remain, results from the PlioMIP2 have shown a widespread model-data agreement (Haywood et al., 2020).

While previous model studies have shown that the high-latitude warming has reduced the equator-pole temperature gradient (Haywood et al., 2013) and weakened tropical circulation such as the Hadley Circulation (Corvec and Fletcher, 2017), the terrestrial warming during the mPWP has been shown to strengthen the WAM and increase the summer rainfall over the Sahel region by more than 1 mm/day (Haywood et al., 2020; Zhang et al., 2016). A similar rainfall increase over Sahel is seen in future projections for both CMIP3 and CMIP5 ensembles, though with a drying located over western Sahel (Roehrig et al., 2013). However, models have been shown to inaccurately capture past rainfall variability and change over West Africa and the Sahel region (Berntell et al., 2018; Roehrig et al., 2013), and there is still little confidence in future projections of the summer rainfall (Biasutti et al., 2008; Cook, 2008; Roehrig et al., 2013). West Africa is a region sensitive to hydrological variability and experienced extended droughts during the 1970s and 1980s (Berntell et al., 2018; Held et al., 2005; Nicholson et al., 2000). There is a large need to increase the confidence in future projections in order to support adaption strategies in the region.

The similarity to modern conditions, as well as the high amount of paleogeological and environmental data from the mPWP, has made it well suited to both evaluate the models' ability to capture a warm climate state and further our understanding of the effects of greenhouse gas forcing and related feedbacks on the global climate system (Haywood et al., 2020; Haywood and Valdes, 2004). In this article we will evaluate the representation of the WAM within the PlioMIP2 ensemble, qualitatively compare it to palaeohydrological reconstructions and discuss the implications for the WAM in a near-future warm climate state with increasing greenhouse gas forcing.

2. Data and methods

2.1. Participating PlioMIP2 models

To examine the behavior of the WAM during the mPWP, data produced by 17 different general circulation models as part of the PlioMIP2 was used (Table 1). Simulations produced within PlioMIP2 are run for at least 500 years (Haywood et al., 2016) towards an equilibrium state, and the last 100 years of the simulations are then used for analysis. In the experimental set-up the $CO_2$ levels are set to 400 ppmv, and the remaining concentrations of trace gases and aerosols are set to pre-industrial levels (Haywood et al., 2016). The simulations are run using standard or enhanced boundary conditions from PRISM4 (Dowsett et al., 2016) as described in Haywood et al., (2016), with

changes to e.g. the topography, bathymetry and land ice cover. All model simulations are run using a mid-Pliocene

land-sea mask except for HadGEM3 and MRI-CGCM 2.3 which use a modern land-sea mask. COSMOS uses

dynamic vegetation (Stepanek et al., 2020), while the remaining 16 models use prescribed vegetation based on

Salzmann et al. (2008). As the models have different horizontal resolutions, the data from the models was bilinearly

interpolated onto a 1° x 1° grid using the software CDO (Climate Data Operators, Schulzweida, 2019) to facilitate

multi-model analysis.

**Table 1: PlioMIP2 models used in this study. Spatial resolution of the atmosphere model indicated by grid cell extent (in degrees longitude x latitude) and number of vertical layers (L).**

| Model ID | Atmospheric resolution | Reference |
|---|---|---|
| CCSM4-NCAR | 1.25 x 0.9, L26 | Feng et al. (2020) |
| CCSM4-Utrecht | 2.5 x 1.9, L26 | |
| CCSM4-UofT | 1.25 x 0.9, L26 | Peltier and Vettoretti (2014); Chandan and Peltier (2017, 2018) |
| CESM1.2 | 1.25 x 0.9, L30 | Feng et al. (2020) |
| CESM2 | 1.25 x 0.9, L32 | Feng et al. (2020) |
| COSMOS | T31 (~3.75 x 3.75), L19 | Stepanek et al. (2020) |
| EC-Earth3-LR | T159 (~1.125 x 1.125), L62 | Zhang et al. (in review) |
| GISS-E2-1-G | 2.0 x 2.5, L40 | Chandler et al. (in prep.) |
| HadCM3 | 2.5 x 3.75, L19 | Hunter et al. (2019) |
| HadGEM3(-GC31-LL) | N96 (~1.875 x 1.25), L85 | Williams et al. (in review) |
| IPSLCM6A-LR | 2.5 x 1.26, L79 | Lurton et al. (2020) |
| IPSLCM5A2 | 3.75 x 1.9, L39 | Tan et al. (2020) |
| IPSLCM5A | 3.75 x 1.9, L39 | Tan et al. (2020) |
| MIROC4m | T42 (~2.8 x 2.8), L20 | Chan and Abe-Ouchi (2020) |
| MRI-CGCM 2.3 | T42 (~2.8 x 2.8), L30 | Kamae et al. (2016) |
| NorESM-L | T31 (~3.75 x 3.75), L26 | Li et al. (2020) |
| NorESM1-F | 1.9 x 2.5, L26 | Li et al. (2020) |

2.3 Methods

The rainfall and West African climate is analyzed over the months July-October (JASO), and the multi-model mean

(MMM) represents the un-weighted average of the PlioMIP2 ensemble. The robustness of the signal is evaluated

using the methodology of Mba et al. (2018), where the signal is considered robust if at least 14 of the 17 models

(=>80%) agree on the sign of the anomaly and the MMM anomaly is equal to or larger than the inter-model standard

deviation. The models are evaluated against their PI-simulation, and the 1901-1930 climatology based on CRU TS

v4 (Climatic Research Unit gridded Time Series; Harris et al., 2020) is included as a reference for the observations.

The seasonal cycle of the WAM is also examined over two sub-regions, Sahel (10-20° N, 20° W-30° E) and the

Coast of Guinea (5-10° N, 20° W-30° E), representing regions characterized by a narrow and a wider or bimodal

rainfall season respectively. A more narrow definition of the Sahel region is also sometimes used (10°W-10°E, e.g. Thorncroft et al., 2011), but our analysis has shown no difference in the seasonal distribution of rainfall compared to a wider region (20°W-30°E). Given the possible northward expansion of the WAM during the mPWP, the seasonal cycle over the Sahara region (20-30°N, 20°W-30°E) is also included.

## 3. Results

### 3.1. Changes in seasonality

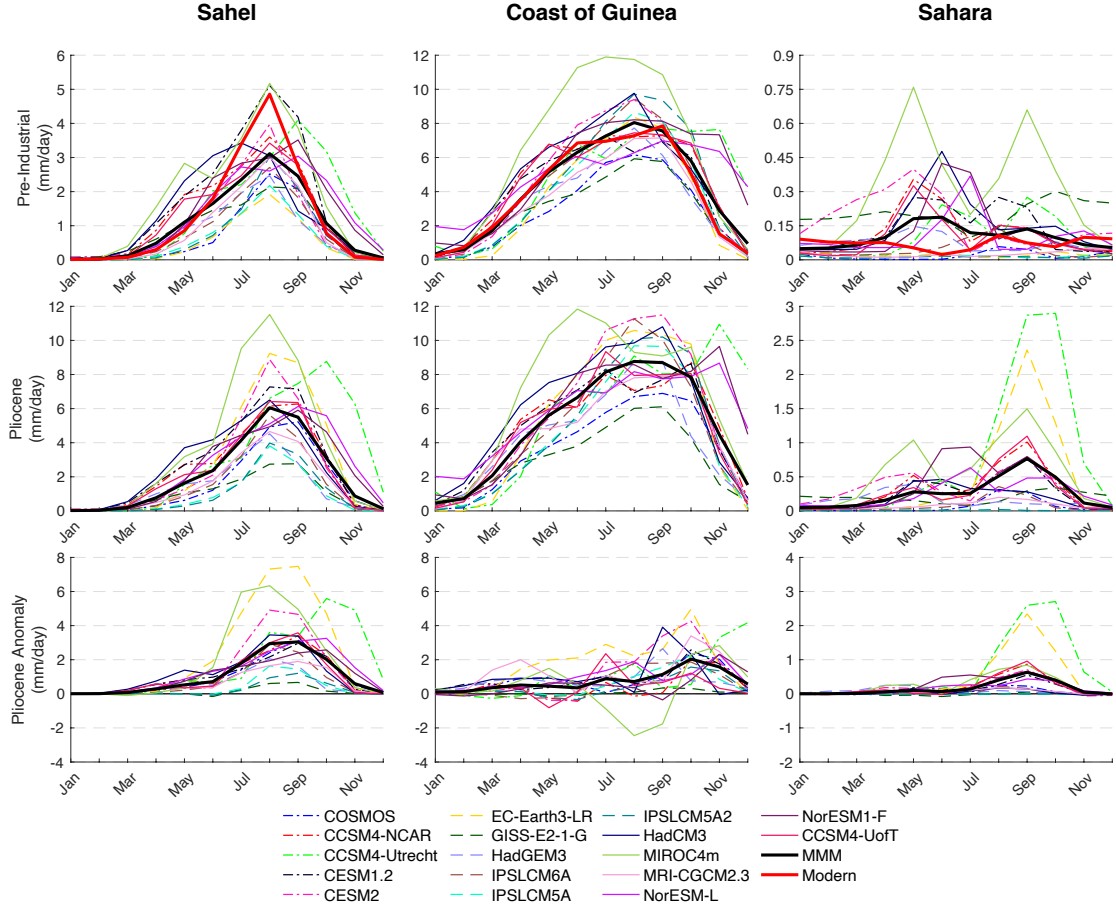

**Fig. 1: Seasonal cycle of rainfall (unit: mm/day) over Sahel (left), the Coast of Guinea (center) and Sahara region (right) for PI (top), mPWP (center) and mPWP anomalies (mPWP-PI, bottom). The multi-model mean MMM (black) is shown together with the individual models, and the modern conditions as derived from observations (Harris et al., 2020) are included as a reference (red).**

The progression of the WAM creates different seasonal cycles of rainfall depending on the region, where northern latitudes in West Africa (south of Sahara) have one clear peak while more southern regions have a wider or bimodal

rainy season (Nicholson et al., 2000). We have therefore divided West Africa into two sub-regions, Sahel (10-20° N, 20° W-30° E) and Coast of Guinea (5-10° N, 20° W-30° E), and shown the results together with the seasonal cycle over the Sahara region (20-30° N, 20° W-30° E). The seasonal cycle of terrestrial rainfall is calculated for each ensemble member and presented together with the MMM for the PI and mPWP simulations separately, as well as for the Pliocene anomaly (mPWP-PI) (Fig. 1). The "modern" seasonal cycle is plotted together with the PI cycle for reference, based on 1901-1930 CRU TS v4 data (Harris et al., 2020).

In agreement with PI observations, the PI MMM shows a seasonal cycle with a peak in rainfall in August over Sahel at 3.1 mm/day. The individual models mainly exhibit the same seasonal cycle; however, four models exhibit highest levels of rainfall shifted to July (HadCM3) or September (CCSM4-Utrecht, NorESM-L and NorESM-F) rather than August. The magnitude of summer rainfall seen in CESM1.2 and MIROC4m is at 5.1 and 5.2 mm/day respectively comparable to modern conditions (4.9 mm/day) (Fig. 1), while the other 15 ensemble members remain within a span of 2-4 mm/day which is considerably below modern levels. The mPWP MMM shows an increase in monsoon rainfall, with the maximum rainfall doubling and reaching 6.1 mm/day in August. The largest increase is shown in EC-Earth3-LR at 7.3 and 7.5 mm/day in August and September, making it reach a maximum of 9.2 mm/day in Pliocene Sahel. As with the PI, the highest level of Pliocene rainfall in the PlioMIP2 ensemble is seen in MIROC4m with 11.5 mm/day in August. All models show an increase in rainfall in the July-October period with the largest increase occurring either in August, September or October, resulting in a lengthening of the WAM. We will therefore base our spatial analysis of the WAM on the July-October (JASO) period, although this does not alter the spatial patterns compared to a shorter monsoon season (July-September).

Over the Coast of Guinea, the PI simulations show higher levels of rainfall through most of the Northern Hemisphere's spring, summer and fall, with the ensemble mean showing a maximum of 8.1 mm/day occurring in August (Fig. 1). This is, both in seasonal distribution and amount, comparable to the PI observations which exhibit maximum rainfall of 7.9 mm/day in September. However, while the observations show a slight bi-modal rainfall distribution, with peaks in June and September, the PI MMM has a wider distribution with a peak in August. 16 of the 17 members have maximum levels of rainfall spanning between 5.9 mm/day and 9.8 mm/day, while MIROC4m again exceeds the remaining models with rainfall reaching 11.9 mm/day in July. The MMM of the mPWP simulations again shows an increase of monsoon rainfall compared to the PI, with positive anomalies throughout the seasonal cycle but showing highest values in October and a secondary peak in July. However, while no

individual models showed negative anomalies during the monsoon season in Sahel, CCSM4-NCAR, MIROC4m and NorESM1-F show decrease in rainfall over the Coast of Guinea in July-September. The remaining models show both increasing and decreasing rainfall during April-June, but mainly positive anomalies from July-November.

The rainfall over the Sahara region remains low for the PI observations and PI simulations, with both PI observations and the PI MMM remaining consistently below 0.2 mm/day throughout the year and 16 of the 17 models staying between 0.0-0.5 mm/day (Fig. 1). MIROC4m again exhibits the highest levels of rainfall, with a peak of 0.8 mm/day in May and a second peak of 0.7 mm/day in September. The mPWP simulations show a clear increase in rainfall over Sahara in the later part of the WAM season, with the mPWP MMM anomalies centered on September and the maximum rainfall reaching 0.8 mm/day. The largest increase is seen in EC-Earth3-LR and CCSM4-Utrecht at 2.3 mm/day in September and 2.6 and 2.7 mm/day in September and October respectively.

## 3.2. Changes in monsoon rainfall

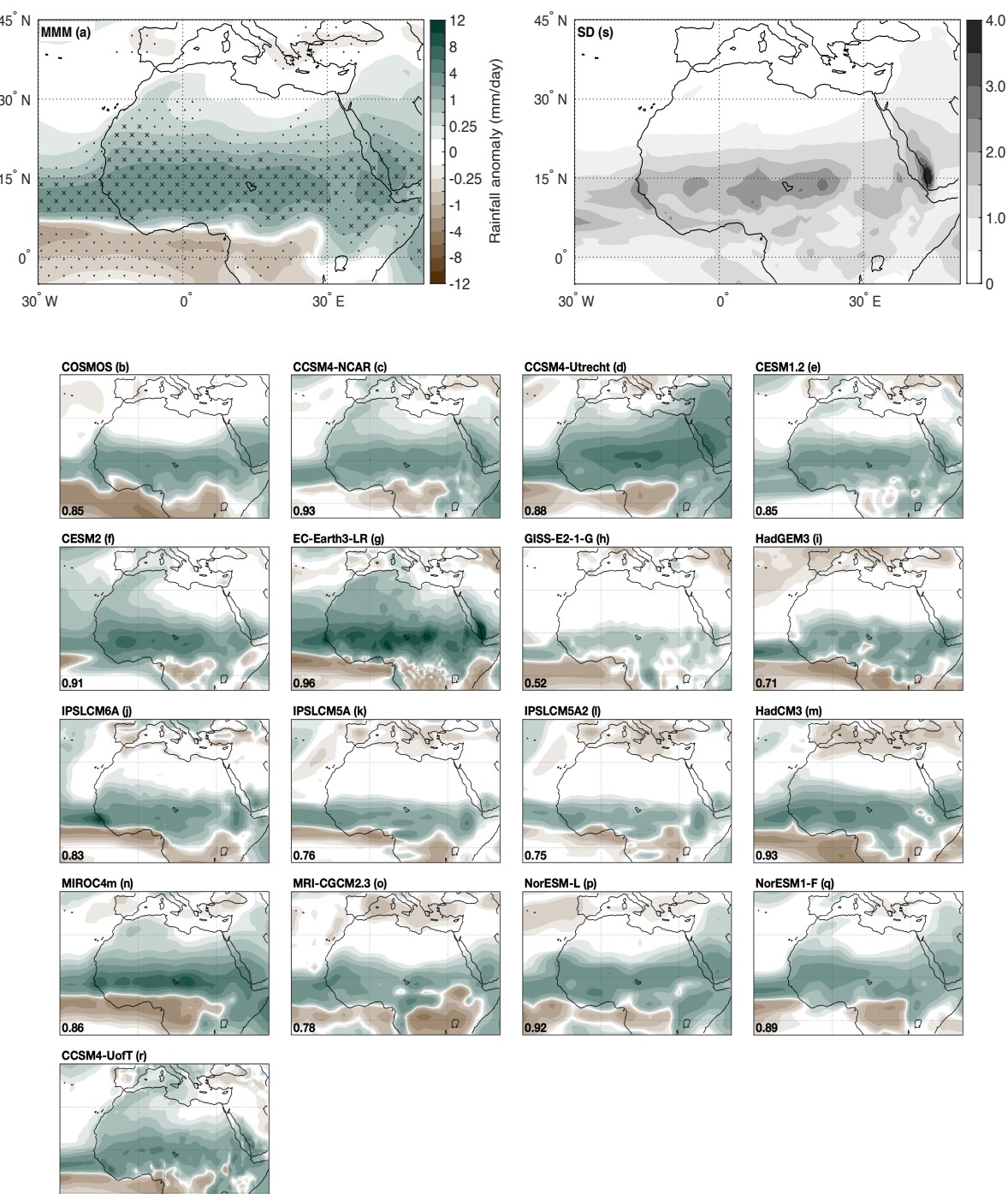

**Fig 2. mPWP July-August-September-October (JASO) rainfall anomalies (mPWP-PI) for the MMM (a) and the individual models (b-r), with all subfigures using the same color bar. Robust signals in (a) are indicated with x, where >80% of the models (14 out of 17) show the same sign of anomaly and the anomaly is equal to or larger than the inter-model standard deviation. Dots indicate that only the first criterion is fulfilled. The pattern correlation between the MMM and individual model is seen in the bottom left corner of (b-r). (s) The inter-model variability is shown as the standard deviation (unit: mm/day).**

To see the changes in the WAM rainfall during the mPWP we look at the JASO rainfall anomalies (mPWP-PI, Fig. 2). The MMM shows a clear dipole pattern with a latitudinal transition at 7°N stretching from the Atlantic Ocean to the eastern part of Northern Africa (Fig. 2a). The robust signal of rainfall increase is centered on Sahel and southern Sahara, covering most of northern Africa and reaching from the Coast of Guinea into northern Sahara. The negative anomalies cover an area stretching from 7°N and continuing south over the Equatorial Atlantic, with the largest decrease located along the Gulf of Guinea.

The large-scale features of the rainfall anomalies are consistent over the individual models, with the rainfall increase centered at 10-15°N and reaching up into southern Sahara, and negative values located over the Gulf of Guinea (Fig. 2a and 2b-r). The results are less consistent along the Coast of Guinea with models indicating slightly different locations of the transition from negative to positive rainfall anomalies. Some models exhibit a rainfall decrease reaching up to 9°N (MIROC4m, GISS-E2-1-G) while other models limit the negative values to only cover the Equatorial Atlantic and Central Africa (CCSM4-UofT, HadCM3). EC-Earth3-LR, CCSM4-UofT, CCSM4-NCAR and HadCM3 show the highest pattern correlation to the MMM at $R=0.96$ (EC-Earth3-LR) and $R=0.93$ (other models) respectively, while GISS-E2-1-G has the lowest correlation ($R=0.52$). The different models show the largest spread over Sahel and southern Sahara (standard deviation of 2-4 mm/day, Fig. 2s). This is a region where all models indicate an increase in rainfall, but the simulated magnitude differs largely, from over 8 mm/day in EC-Earth3-LR and MIROC4m to around 1 mm/day for GISS-E2-1-G and IPSLCM5A2. A spatial mean of the rainfall anomalies over Sahel (Fig. 3) shows a similar spread, with the highest values for EC-Earth3-LR and MIROC4m (6.1 and 5.0 mm/day) and the lowest for GISS-E2-1-G and IPSLCM5A2 (0.4 and 0.7 mm/day). The remaining 13 models all show an increase of 1-4 mm/day over Sahel with a MMM of 2.5 mm/day.

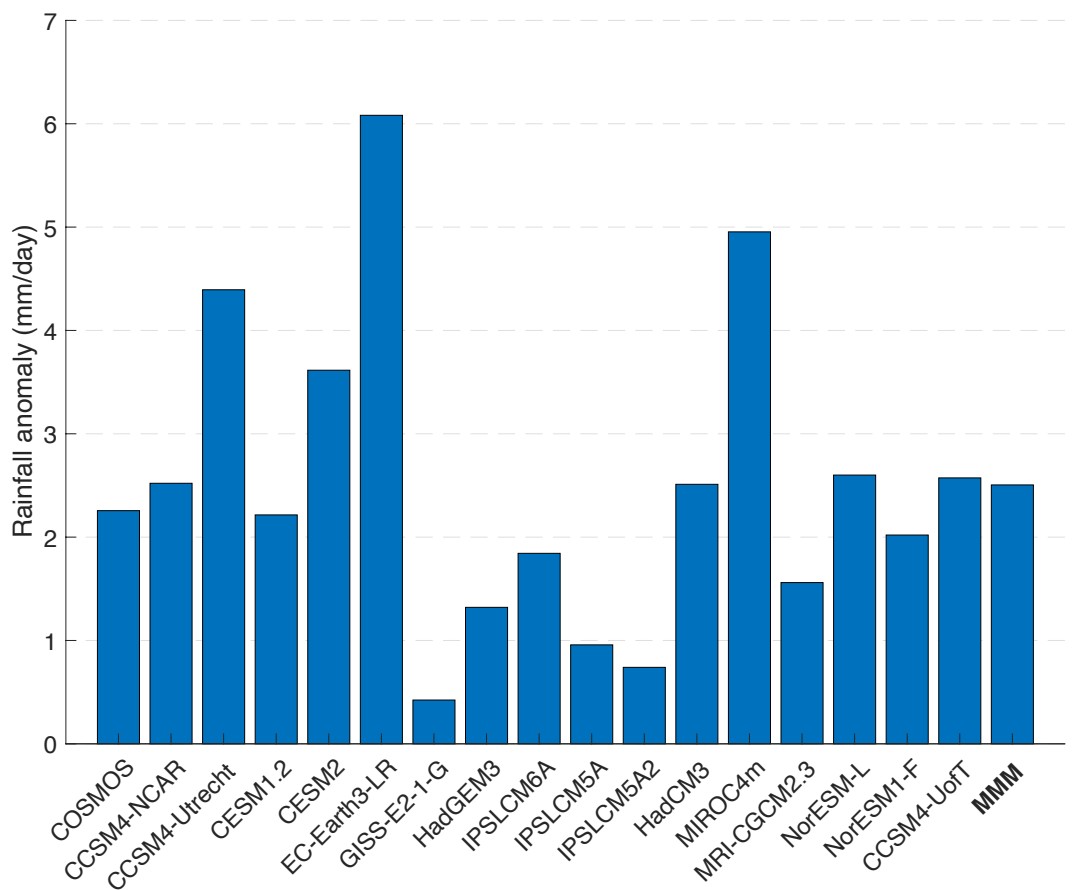

**Fig. 3. Mean July-October (JASO) Sahel (10-20°N, 20°W-30°E) mPWP rainfall anomaly (mPWP-PI, unit: mm/day) for the individual PlioMIP2 ensemble models, together with the MMM.**

Looking at the latitudinal mean JASO rainfall (Fig. 4) we can also see that the rainbelt, i.e. the latitudinal band of maximum rainfall during the WAM, has shifted northward in the mPWP and is centered at 9.5°N with the largest rainfall increase of 2.0 - 3.1 mm/day occurring between 9.5°N and 17.5°N for the MMM. The ensemble does however still exhibit a large spread, with three models showing a maximum increase to the south of the MMM (IPSLCM5A, IPSLCM5A2 and IPSLCM6A), four models to the north (CCSM4-Utrecht, CESM1.2, NorESM-L

and NorESM1-F) and two models showing a substantially larger increase than the MMM (EC-Earth3-LR and MIROC4m).

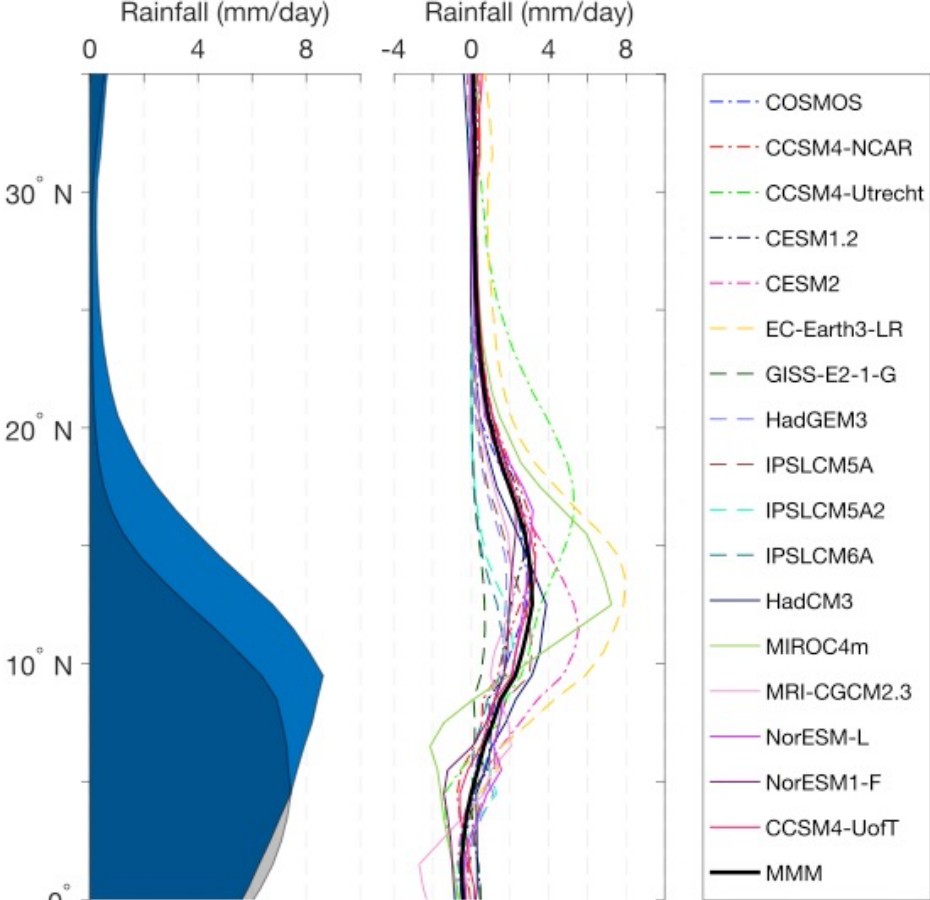

**Fig. 4. (left) Latitudinal mean terrestrial rainfall for MMM PI (grey) and mPWP (blue), with dark blue where they overlap, and (right) latitudinal mean July-October (JASO) rainfall anomalies (mPWP-PI) for the individual models and for the MMM.**

3.3. The dynamics for the changes in WAM rainfall

To understand the dynamics behind the increased rainfall in West Africa during the mPWP, the sea level pressure, horizontal wind at 850 hPa and near surface temperature anomalies (mPWP-PI) are analyzed for each individual model.

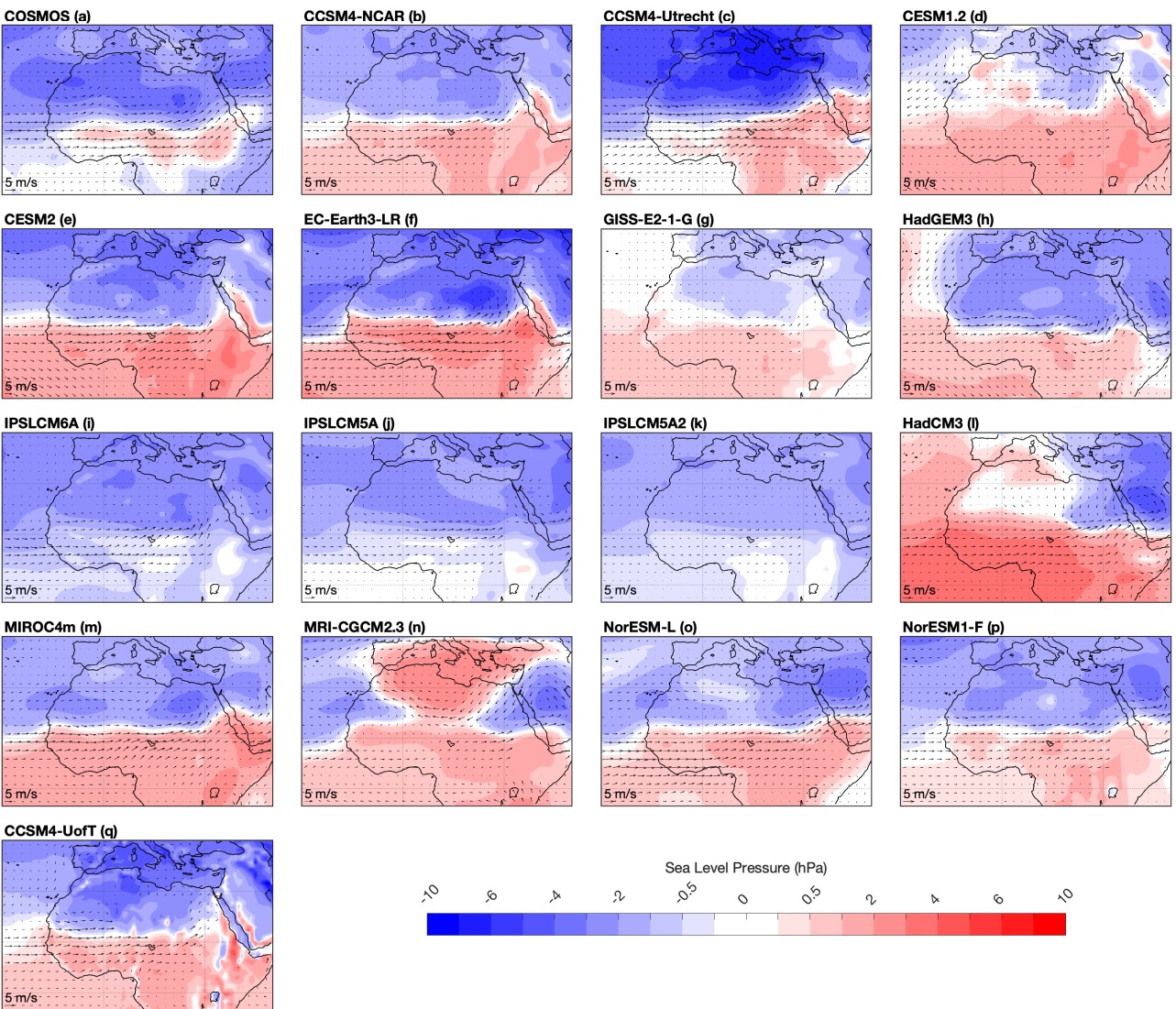

**Fig. 5. July-October (JASO) mean sea level pressure (shading) and 850 hPa horizontal wind (vectors) anomalies for the PlioMIP2 ensemble members (a-q).**

Sea level pressure anomalies for the monsoon season (JASO, mPWP-PI) are shown in Fig. 5 for the individual PlioMIP2 models. All models except MRI-CGCM 2.3 (Fig. 5n) show a deepening of the low-pressure area across the Sahara region (negative anomalies) and a strengthening of the negative latitudinal pressure gradients between Sahara and the Equatorial Atlantic. CCSM4-NCAR, EC-Earth3-LR and CCSM4-UofT (Fig. 5f and 5q), the models with some of the highest pattern correlation in rainfall to the ensemble mean, all exhibit a clear north/south dipole pattern with negative sea level pressure anomalies over Sahara continuing northward into Europe, and positive anomalies over Sahel, the Coast of Guinea and the Equatorial Atlantic. The same dipole pattern, with a latitudinal transition at approx. 17°N, is also seen in seven additional ensemble members (CCSM4-Utrecht, CESM2, GISS-E2-1-G, HadGEM3, MIROC4m, NorESM-L and NorESM1-F), but while MRI-CGCM 2.3 exhibits positive sea

level pressure anomalies south of 15°N, the negative anomalies over Sahara are divided by positive anomalies over northern Africa and southern Europe, centered on the Mediterranean region, resulting in a quadrupole-type pattern (Fig. 5n). The three IPSL models (IPSLCM6A, IPSLCM5A and IPSLCM5A2) show negative anomalies or weak positive anomalies south of 17°N, forming a weaker enhancement of the latitudinal pressure gradient relative to the other PlioMIP2 models.

Associated with the deepening of the Saharan Heat Low and strengthening of the latitudinal pressure gradients is an anomalous cyclonic flow and strengthened westerly/southwesterly horizontal winds at the 850 hPa level, going from the Equatorial Atlantic into Sahel and Sahara (Fig. 5 a-q). This is seen in all models, although at different magnitudes, with the highest increase in wind speed seen in CCSM4-Utrecht, EC-Earth3-LR and MIROC4m (Fig. 5c, 5f and 5m), and the lowest increase for GISS-E2-1-G, IPSLCM5A and IPSLCM5A2 (Fig. 5g, 5j and 5k).

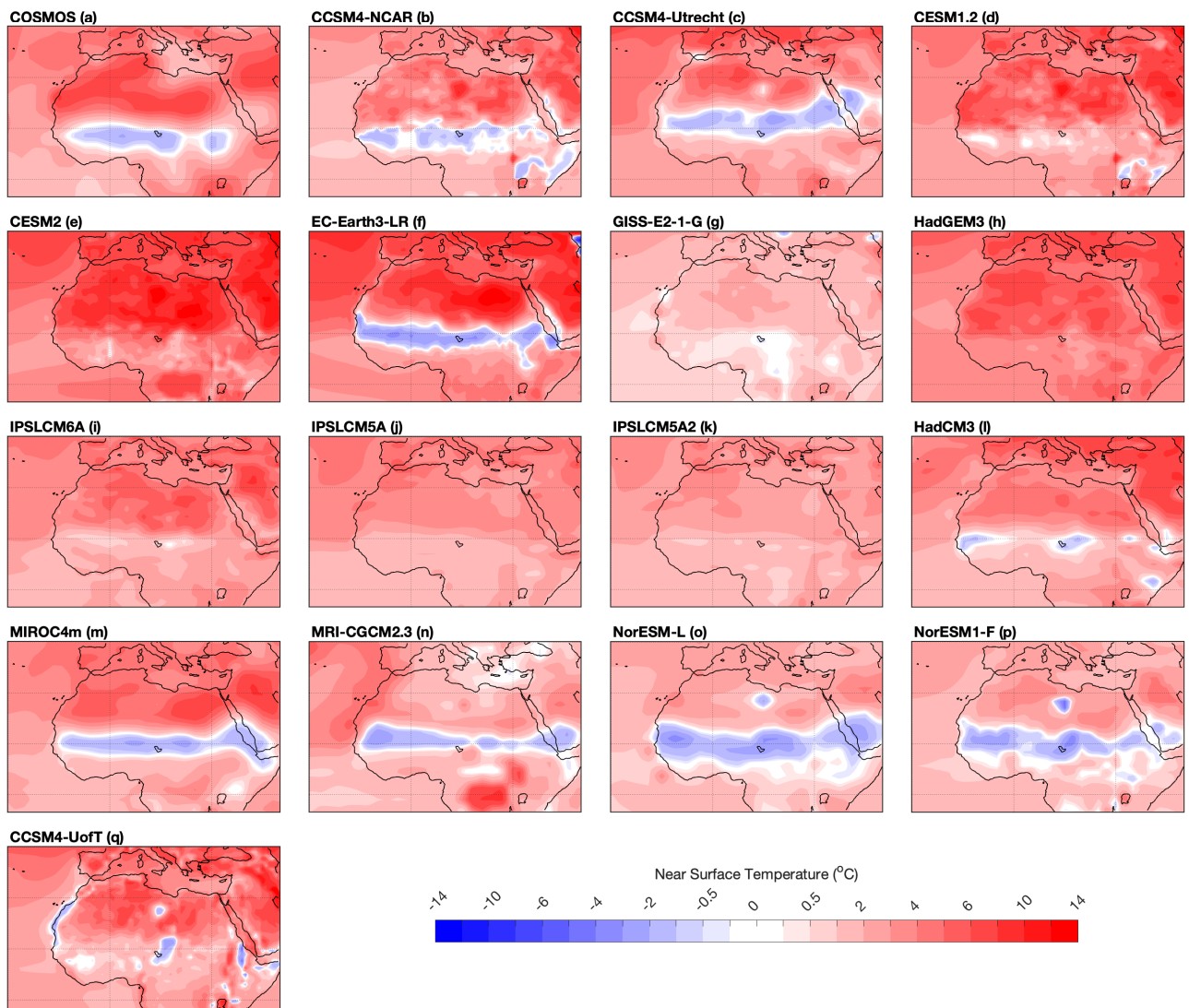

**Fig. 6. July-October (JASO) mean near surface temperature anomalies (ΔSAT, mPWP-PI) for the PlioMIP2 ensemble members (a-q).**

The JASO near surface temperature anomalies (ΔSAT, mPWP-PI, Fig. 6) shows a strengthened north-south temperature gradient between the Sahara Desert and the Equatorial Atlantic for all models except MRI-CGCM 2.3 (Fig. 6n). The temperature increase either stretches relatively uniformly across Sahara as in EC-Earth3-LR, COSMOS and CCSM4-UofT, or exhibits two separate centers, one in Western Sahara and one in Eastern Sahara, as for MIROC4m, NorESM-L and NorESM1-F. MRI-CGCM 2.3 (Fig. 6n) has positive temperature anomalies located mainly outside Sahara, both centered along the western coast of Sahara and over eastern Sahara and the Arabian peninsula. An area of negative temperature anomalies is located over the Mediterranean region, and its surrounding areas in northern Sahara exhibit a weaker warming than the neighboring areas of the Sahara Desert. Nine models show clear latitudinal bands of negative anomalies stretching across northern Africa at approx. 15°N

(COSMOS, CCSM4-NCAR, CCSM4-Utrecht, EC-Earth 3-LR, HadCM3, MIROC4m, MRI-CGCM 2.3, NorESM-L, NorESM1-F), similar to the latitude of maximum rainfall increase. CCSM4-UofT temperatures exhibit negative anomalies more dispersed over northern Africa, located mainly along the western coastline of Sahara and over the central Sahel region.

4. Discussion

4.1 The paleo-proxy evidence for WAM during the mid-Pliocene

The mPWP is often used as an analog for near-future climate change due to its similar-to-modern paleogeography and high concentrations of $CO_2$ in the atmosphere (Corvec and Fletcher, 2017; Dowsett et al., 2013; Sun et al., 2013), and both marine and terrestrial proxy reconstructions indicate a climate with higher sea surface and surface air temperatures than present (Dowsett et al., 2013; Salzmann et al., 2008). Model/data comparison using PlioMIP1

indicated that the models underestimated the high latitude warming by up to 15 °C while overestimating the low latitude temperatures by 1-6 °C (Dowsett et al., 2013; Haywood et al., 2013; Salzmann et al., 2013). A comparison of atmosphere-only general circulation models (AGCM) and coupled ocean-atmosphere models (AOGCM) showed that AGCMs, using prescribed SSTs based on paleo reconstructions, produce a much stronger WAM compared to models using a coupled ocean-atmosphere configuration, believed to be due to the overestimation of SST and SAT

in the tropics in the PlioMIP1 AOGCM's (Zhang et al., 2016). Analysis of the PlioMIP2 ensemble by Haywood et al. (2020) indicates a widespread model/data agreement for SSTs and little systematic temperature bias in the tropics, suggesting a reduced underestimation of the WAM in the PlioMIP2, but the relatively low availability of palaeohydrological proxies covering West Africa makes it difficult to perform a similar model/data comparison for the WAM and its related rainfall (Salzmann et al., 2008, 2013). However, several studies of proxy reconstructions

across Northern Africa indicates a more humid climate during the mid-Pliocene. Palynological data records suggest a higher density of tree cover and an expansion of woodland and savanna in Northern Africa at the expense of deserts (Bonnefille, 2010; Salzmann et al., 2008). Multi-proxy studies analyzing e.g. plant-wax and dust records in marine sediment cores taken off-shore of West Africa indicate wetter conditions during the mid-Pliocene (deMenocal, 2004; Feakins et al., 2005; Kuechler et al., 2018), which is qualitatively consistent with the results

from the PlioMIP2 ensemble (Fig. 2). The expansion of forest into the Sahara region is also seen in the results from COSMOS (Stepanek et al., 2020), which is the only member of the PlioMIP2 ensemble that is run with dynamic

vegetation. It is also important to note that the PlioMIP2 ensemble is designed to simulate the MIS KM5c within the mPWP (Haywood et al., 2020, 2016), and while it represents a useful comparison to modern conditions it might not represent the full climate variability within the mPWP, possibly affecting model-data comparisons (Prescott et al., 2014; Samakinwa et al., 2020).


## 4.2 WAM – PI and mid-Pliocene

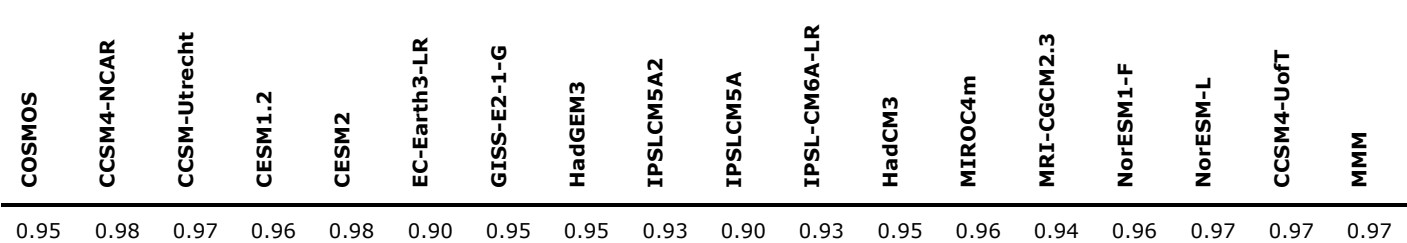

| COSMOS | CCSM4-NCAR | CCSM-Utrecht | CESM1.2 | CESM2 | EC-Earth3-LR | GISS-E2-1-G | HadGEM3 | IPSLCM5A2 | IPSLCM5A | IPSL-CM6A-LR | HadCM3 | MIROC4m | MRI-CGCM2.3 | NorESM1-F | NorESM-L | CCSM4-UofT | MMM |
|---|---|---|---|---|---|---|---|---|---|---|---|---|---|---|---|---|---|
| 0.95 | 0.98 | 0.97 | 0.96 | 0.98 | 0.90 | 0.95 | 0.95 | 0.93 | 0.90 | 0.93 | 0.95 | 0.96 | 0.94 | 0.96 | 0.97 | 0.97 | 0.97 |

**Table 2. Pattern correlation of July-September mean rainfall over West Africa (0-25° N, 30° W-30° E) between PlioMIP2 PI simulations, including the MMM, and observational data (CRU TS v4.: 1901-1930 mean).**

High pattern correlations of JAS rainfall over West Africa ($R$>0.90; Table 2) between the PI simulations and
climatologies based on observational data (CRU: 1901-1930 (Harris et al., 2020)) for all models ($R$=0.97 for MMM) indicate that the models are able to sufficiently reproduce the WAM rainfall pattern. However, looking at the absolute values (Fig. 1) it is clear that while they capture the general seasonal cycle with rainfall peaking in July-September, most models still underestimate the magnitude of the modern summer rainfall over Sahel by 1-3 mm/day, the only exceptions being CESM1.2 and MIROC4m with >5 mm/day of rainfall in August. This is
consistent with our general understanding that models struggle to capture West African rainfall (e.g. Roehrig et al., 2013).

The MMM shows a clear increase in summer rainfall in the Sahel region, and to a lesser extent also over the Sahara region, consistent with a strengthened WAM during the mPWP (Fig. 1). The anomalies are centered on mid to late summer (August-September), which indicates a later withdrawal of the WAM and a lengthened monsoon season.
The monsoonal rainfall over the (terrestrial) Coast of Guinea also exhibits larger positive anomalies over the later months of the summer rainfall, further suggesting an intensification of the WAM rainfall towards the end of the monsoon season as well as a later withdrawal during the mid-Pliocene.

There is a large consistency within the ensemble regarding the general features of the mPWP WAM (Fig. 2). All models are showing a JASO rainfall increase over Sahel reaching up into Sahara, and negative anomalies over the

Equatorial Atlantic, indicating an intensification and northward shift as well as expansion of the WAM. The changes are statistically robust and consistent with previous studies on both PlioMIP1 and 2 where the tropics, particularly the Northern Hemisphere monsoon region, is identified as a region with a robust rainfall signal during the mid-Pliocene (Haywood et al., 2020; Li et al., 2018; Pontes et al., 2020; Zhang et al., 2016). A model/proxy comparison by Feng et al. (in review, Sci. Adv.) has also shown that the wetter conditions seen over Sahel and West Africa in PlioMIP2 (Fig. 2) are consistent with the available qualitative indicators of mPWP hydroclimate, although the magnitude of change cannot be obtained from these proxy datasets and therefore not compared quantitatively to the PlioMIP2 results.

The signal is markedly stronger in the PlioMIP2 compared to PlioMIP1, where the MMM shows a doubling of the rainfall increase over Sahel from 1-2 mm/day in PlioMIP1 (Zhang et al., 2016) to 2-4 mm/day in PlioMIP2 (Fig. 2). Note that the use of June-August as the monsoon season in Zhang et al. (2016) might also have contributed to the discrepancy, especially given the rainfall increase seen in our results over the later part of the monsoon season (Fig 1). The updated boundary conditions from PRISM3 to PRISM4 might have contributed to this enhancement, where a sensitivity study by (Samakinwa et al., 2020) using COSMOS has shown that the updated paleogeography played the largest role in the changes to the global large-scale climate between PlioMIP1 and PlioMIP2. However, these changes appear to be more pronounced in high than low latitudes (Samakinwa et al., 2020), and HadGEM3 and MRI-CGCM 2.3, which did not implement the enhanced boundary conditions, still exhibits a precipitation response over West Africa within 1 standard deviation of the MMM (Fig. 2). Haywood et al. (2020) instead suggest that the sensitivity of the individual ensemble members to the mid-Pliocene boundary conditions is mostly related to the model parameterization and initial conditions, with model improvements between the two phases also playing a role. Within model families, later model versions, run with the same boundary and initial conditions, tend to be more sensitive than earlier versions (Haywood et al., 2020). This is also consistent with our results, where e.g., CESM2 exhibits larger rainfall anomalies than CESM1.2 and IPSLCM6A exhibits larger rainfall anomalies than IPSLCM5A and IPSLCM5A2 (Fig. 3).

The precipitation response over West Africa of the models is also in many ways similar to their global response, where e.g. the weakest rainfall increase in Sahel is seen in GISS-E2-1-G (Fig 2), consistent with the model's low global rainfall response to the mid-Pliocene boundary conditions (Haywood et al., 2020). Models which were identified as having a larger land/sea rainfall anomaly contrast, with a larger rainfall enhancement over land

compared to the ocean (Haywood et al., 2020), are also the models which show a larger rainfall increase in Sahel (EC-Earth3-LR, HadCM3, MIROC4m, NorESM1-F, NorESM-L and CCSM4-UofT). However, COSMOS, which did not show a clear land rainfall enhancement globally, exhibits similarly strong levels of rainfall increase in Sahel, and even slightly more than NorESM1-F (2.3 and 2.0 mm/day respectively).

Haywood et al. (2020) also suggests that, in general, models exhibiting large SAT sensitivity (i.e., high global mean ΔSAT) also exhibit a larger rainfall change (globally), but there is still uncertainty in changes in more regional patterns. While this finding is consistent with the results from EC-Earth3-LR, which has both one of the highest increase in Sahel rainfall and global SAT (De Nooijer et al., 2020), there is less consistency within the remaining ensemble. MIROC4m and IPSLCM6A both exhibit similar global ΔSAT (De Nooijer et al., 2020), but their rainfall change differs by close to a factor of 3 (Fig. 3). The PlioMIP2 models however show a consistent JASO warming of the Sahara Desert (Fig. 6), and if the analysis is limited to the Sahara (10°W-10°E, 20-30°N) a clear link between the ΔSAT and the rainfall increase can be observed (Fig. 7, $R$=0.42, 90% significance, if only looking at JAS: $R$=0.50, 95% significance). The warming of the Sahara Desert and strengthened latitudinal temperature gradient between the Sahara region and the equatorial Atlantic leads to a deepening of the thermally induced Saharan Heat Low (Fig. 5) (Lavaysse et al., 2009). This deepened Saharan Heat Low induces low-level convergence and strengthens the southwesterly flow, bringing moisture from the equatorial Atlantic into the continent, leading to increased moisture availability and rainfall over Sahel and parts of Sahara, and in summary indicating a strengthened WAM (Fig. 5).

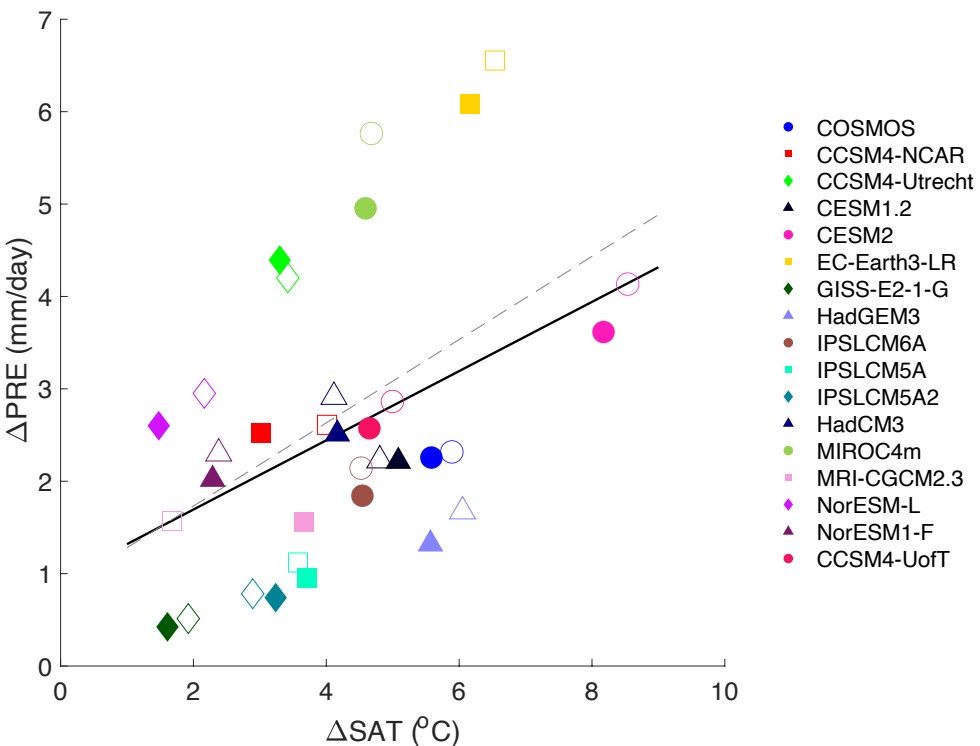

**Fig. 7. Sahel (10-20°N, 20°W-30°E) rainfall and Sahara (20-30°N, 10°W-10°E) near-surface temperature anomalies (mPWP-PI). Filled markers and black line show JASO anomalies and least-square fit, while non-filled markers and dashed line show JAS anomalies and least-square fit.**

4.3 Role of mid-Pliocene forcing and boundary conditions

The warming of the Sahara region and subsequent strengthening of the WAM is similar to what we see during other warm climates, such as the Mid-Holocene and Last Interglacial period (Gaetani et al., 2017; Otto-Bliesner et al., 2020), but given the boundary conditions in the mid-Pliocene simulations this warming over Sahara is most likely driven by the changes in the atmospheric $CO_2$-concentration, topography and related vegetation changes over West Africa. Studies of model simulations as well as observational data has shown that greenhouse gas forcing leads to a land-ocean warming contrast, with a larger temperature increase over land (Byrne and O'Gorman, 2013; Haywood et al., 2020; Lambert et al., 2011). The contrast is a result of the lower moisture availability over land influencing the lapse rate and leading to a higher warming compared to the ocean (Byrne and O'Gorman, 2013), which is consistent with the strong response over the arid Sahara region (Fig. 6). Studies show that this land/ocean warming contrast is present in both equilibrium and transient simulations (Lambert et al., 2011), and future scenarios of climate change show a continued land/ocean contrast and warming of the Sahara region (Boer, 2011; Sutton et al., 2007), leading to strengthened latitudinal temperature gradients.

In addition to the $CO_2$ forcing, the majority of the PlioMIP2 ensemble members were forced by the PRISM4 boundary conditions, with reconstructed distributions of topography, bathymetry, land ice cover, vegetation and land/sea mask being amongst the major changes compared to modern geography (Dowsett et al., 2016; Haywood et al., 2016). While the PlioMIP2 set-up, with orography and vegetation being changed together, makes it difficult to distinguish between the impact of the two boundary conditions, the sensitivity simulations by COSMOS indicate that the mid-Pliocene paleogeography still contributes to the increased rainfall over West Africa in mPWP (Stepanek et al., 2020). The enhanced topography with closed Arctic gateways has been shown to strengthen the Atlantic Meridional Overturing Circulation (AMOC), contributing to the warming in the North Atlantic seen in PlioMIP2 as well as the reduced model-proxy mismatch between PlioMIP1 and 2 (Zhang et al., 2020). However, although the strength of the AMOC has been linked to rainfall variability over Sahel (Mulitza et al., 2008), HadGEM3, which did not include the enhanced topography and instead exhibit a weakening of the AMOC (Zhang et al., 2020), still exhibit a rainfall increase over West Africa close to (within 1 standard deviation of) the MMM (Fig. 2). These findings indicate that other mid-Pliocene boundary conditions remain important related to rainfall changes over West Africa.

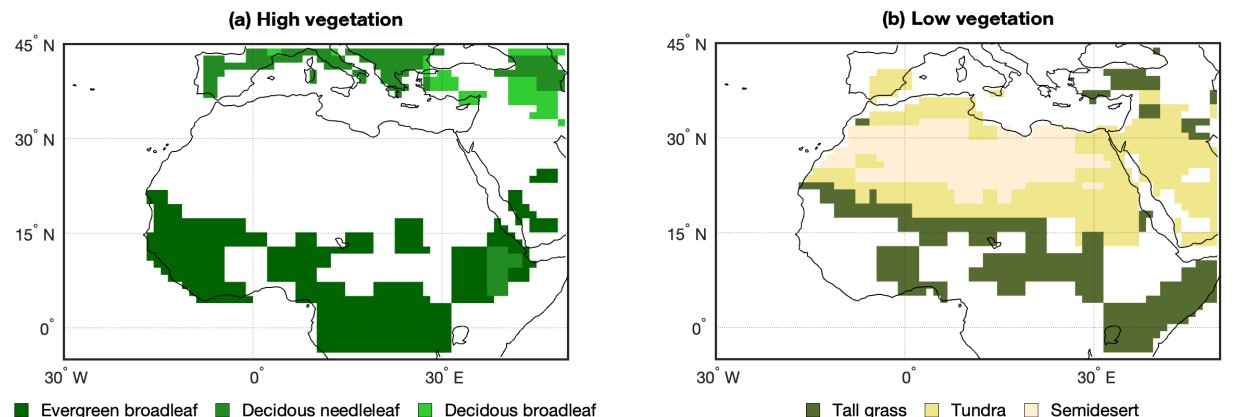

**Fig. 8. Prescribed PRISM4 (a) high and (b) low vegetation used as boundary conditions for PlioMIP2 simulations** (Dowsett et al., 2016), **sourced from** Salzmann et al. (2008)**.**

Land surface changes are also known to impact rainfall over West Africa, where, e.g., expansion of vegetation into the Sahara region at the expense of desert leads to a decrease of the surface albedo and an increase in equivalent potential temperature, further strengthening the Sahara Heat Low, and subsequently the WAM, leading to a vegetation-albedo feedback (Charney, 1975). Additionally, later modelling studies have emphasized the role of soil moisture (Patricola and Cook, 2008) and evapotranspiration (Rachmayani et al., 2015) in the vegetation-precipitation feedback due to their effect on low-level moist static energy, convective instability and surface latent

heat flux anomalies. These feedback mechanisms have been shown to strengthen the response of the WAM to external forcing in other past climates (e.g., Braconnot et al., 1999; Chandan and Peltier, 2020; Claussen and Gayler, 1997; Messori et al., 2019; Rachmayani et al., 2015), and the enhanced vegetation in the PlioMIP2 ensemble (Haywood et al., 2020; Salzmann et al., 2008) is likely to have contributed to the strengthening of the mPWP WAM and West African summer rainfall. While vegetation and orography are changed together in PlioMIP2, making it difficult to separate the impact of the two boundary conditions, sensitivity experiments indicate that they play a large role in the mPWP rainfall increase over West Africa (Chan and Abe-Ouchi, 2020; Hunter et al., 2019; Kamae et al., 2016). Studies by Feng et al. (in review, Sci. Adv.) further indicate that the enhanced vegetation drives the mPWP hydroclimatological changes over Sahel to a larger extent than the enhanced topography. In PlioMIP2 the enhanced vegetation is used as a boundary condition (Fig. 8), but a northward expansion of vegetation is also seen in the dynamic vegetation model COSMOS (Stepanek et al., 2020, Fig. 25 e,f), indicating that the mPWP rainfall increase over West Africa and Sahel of 2.5 mm/day for the MMM and 2.3 mm/day for COSMOS (Fig. 3) can support the enhanced vegetation cover seen in the PRISM4.

As the latitudinal land-ocean temperature gradient is central to the development and strength of the WAM through the development of the Saharan Heat Low (Lavaysse et al., 2009), the results have strong implications for future scenarios. Unlike the results in PlioMIP2, and previously in PlioMIP1 (Zhang et al., 2016), which exhibit a uniform rainfall increase over West Africa, both CMIP3 (SRES A2) and CMIP5 (RCP8.5) model ensembles show a drying over western Sahel and a rainfall increase over central and eastern Sahel (Roehrig et al., 2013). As analysis of both CMIP3 and CMIP5 ensembles show a large spread in projected rainfall change in the Sahel region which weakens its confidence in future projections (Roehrig et al., 2013), our results support a future strengthening of the WAM and rainfall increase over West Africa and Sahel in a high $CO_2$ scenario. However, given the role of the enhanced vegetation in the strengthening of the mPWP WAM, this will also depend on the future vegetation changes in the region which still remain elusive (Bathiany et al., 2014).

5. Conclusion

The PlioMIP2 ensemble shows a clear rainfall increase over West Africa, with the largest increase located over Sahel, and a strengthening of the WAM leading to the rainfall reaching farther in over the continent. These results are consistent with geological evidence which suggests a more humid climate during the mid-Pliocene (Kuechler et al., 2018; Salzmann et al., 2008). Some regional differences occur among the ensemble members, mainly along

the Coast of the Gulf of Guinea where some models indicate drier conditions while other models indicate a rainfall increase. The largest inter-model variability is centered along Sahel, where the magnitude of the rainfall increase varies largely between the models. The strengthened WAM is driven by the warming of the Sahara region and subsequent deepening of the Saharan Heat Low, most likely due to the greenhouse gas forcing, vegetation changes and land/ocean warming contrast. The deepened Saharan Heat Low leads to anomalous cyclonic flow and increased moisture flux into the Sahel region, resulting in a northward shift and intensification of the rainbelt. Given the potential for using the PlioMIP2 as an analogue for near-future scenarios, these results suggest a more uniform rainfall increase over West Africa and the Sahel region, unlike the east-west contrast seen in both CMIP3 and CMIP5 future projections (Roehrig et al., 2013). Alternatively, these results suggest that the extent of analogism between mid-Pliocene and future climate in the context of rainfall over West Africa may depend on the long-term response of vegetation to the $CO_2$ forcing and on the speed with which climate adapts to future carbon dioxide burden - CMIP simulations of a transient climate and mid-Pliocene simulations of a quasi-equilibrium climate representing endmembers of potential future conditions.

*Data availability.* The model data can be downloaded from PlioMIP2 data server located at the School of Earth and Environment of the University of Leeds; an email can be sent to Alan Haywood (a.m.haywood@leeds.ac.uk) for access.

*Author contributions.* Ellen Berntell and Qiong Zhang designed the work, Ellen Berntell did the analysis and wrote the manuscript. All co-authors provided the PlioMIP2 model data and commented on the manuscript.

*Competing interests.* The authors declare that they have no conflict of interest.

*Acknowledgements.* This research has been supported by the Swedish Research Council (Vetenskapsrådet, grant no. 2013-06476 and 2017-04232). The model simulations with EC-Earth3 and data analysis were performed by resources provided by ECMWF's computing and archive facilities and the Swedish National Infrastructure for Computing (SNIC) at the National Supercomputer Centre (NSC) partially funded by the Swedish Research Council through grant agreement no. 2018-05973.

GL and CS acknowledge computational resources from the Computing and Data Centre of the Alfred-Wegener-Institute – Helmholtz-Centre for Polar and Marine Research. GL and CS acknowledge funding via the Helmholtz Climate Initiative REKLIM and the Alfred Wegener Institute's research programme "Changing Earth - Sustaining our Future".

Bette L. Otto-Bliesner, Esther C. Brady and Ran Feng acknowledge that material for their participation is based upon work supported by the National Center for Atmospheric Research, which is a major facility sponsored by the National Science Foundation (NSF) (cooperative agreement no. 1852977 and NSF OPP grant no. 1418411). Ran Feng is also supported by NSF grant no. 1903650. The CESM project is supported primarily by the National Science Foundation. Computing and data storage resources, including the Cheyenne supercomputer (https://doi.org/10.5065/D6RX99HX), were provided by the Computational and Information Systems Laboratory (CISL) at NCAR. NCAR is sponsored by the National Science Foundation.

WLC and AAO acknowledge funding from JSPS (KAKENHI grant no. 17H06104 and MEXT KAKENHI grant no. 17H06323) and computational resources from the Earth Simulator at JAMSTEC, Yokohama, Japan.

The NorESM simulations benefitted from resources provided by UNINETT Sigma2 – the National Infrastructure for High Performance Computing and Data Storage in Norway

WRP and DC were supported by Canadian NSERC Discovery Grant A9627, and they wish to acknowledge the support of SciNet HPC Consortium for providing computing facilities. SciNet is funded by the Canada Foundation for Innovation under the auspices of Compute Canada, the Government of Ontario, the Ontario Research Fund – Research Excellence, and the University of Toronto.

CJRW and DJL acknowledge the financial support of the UK Natural Environment Research Council (NERC)-funded
SWEET project, research grant NE/P01903X/1.

Development of GISS-E2.1 was supported by the NASA Modeling, Analysis, and Prediction (MAP) Program. CMIP6 simulations with GISS-E2.1 were made possible by the NASA High-End Computing (HEC) Program through the NASA Center for Climate Simulation (NCCS) at Goddard Space Flight Center.

The PRISM4 reconstruction and boundary conditions used in PlioMIP2 were funded by the U.S. Geological Survey Climate
and Land Use Change Research and Development Program. Any use of trade, firm, or product names is for descriptive purposes only and does not imply endorsement by the U.S. Government.

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
