# Peer review of "Mid-Pliocene West African Monsoon Rainfall as simulated in the PlioMIP2 ensemble"

_Climate of the Past, 2021_

## Author Response (AR1)

**Author Response**

On the manuscript

**Mid-Pliocene West African Monsoon Rainfall as simulated in the PlioMIP2 ensemble**

by Ellen Berntell et al., submitted to *Climate of the Past* (https://doi.org/10.5194/cp-2021-16).

**We thank the reviewers for their valuable and constructive comments and the time and effort spent reviewing our manuscript. Below is listed our response to both the general and specific questions, with the reviewer's comments in black and our replies and revised text in blue.**
* * *
**Author response to Reviewer #1**

**Reviewer Comment #1:** The main explanation for the increase in monsoon precipitation is seen in the increased atmospheric CO2 level in this study. However, the modern atmospheric CO2 content is at the same level and observations show a precipitation band in North Africa that does not extend very far north and also nowhere near as high precipitation rates as shown in this study for the mid-Pliocene. The question therefore arises as to what extent other causes might be (at least partly) responsible for the area-wide increase in precipitation. It would be helpful to introduce and discuss other boundary conditions that may affect the monsoon rainfall. Land surface changes are only briefly addressed. These certainly contribute to an increase in precipitation, but are there other factors? In this context, I would suggest showing a map with the prescribed vegetation pattern and maybe also a map for the vegetation anomaly simulated by COSMOS. And I recommend to discuss the vegetation influence in more detail.

**Response:** Thank you for this comment. We agree with the reviewer that the higher atmospheric CO2 concentration is indeed not the only boundary condition affecting the precipitation in North Africa during the Pliocene. Vegetation/land surface changes are known to impact the precipitation, and we have expanded our discussion on the role of non-CO2 forcing (vegetation, topography, etc.). As per your suggestion, we have included a map of the prescribed mid-Pliocene vegetation and discussed it in relation to the precipitation changes as well as the vegetation patterns simulated by the dynamic vegetation model COSMOS (L. 367-397).

> **Revised:** In addition to the CO2 forcing, the majority of the PlioMIP2 ensemble members were forced by the PRISM4 boundary conditions, with reconstructed distributions of topography, bathymetry, land ice cover, vegetation and land/sea mask being amongst the major changes compared to modern geography (Dowsett et al., 2016; Haywood et al., 2016). While the PlioMIP2 set-up, with orography and vegetation being changed together, makes it difficult to distinguish between the impact of the two boundary conditions, the sensitivity simulations by COSMOS indicate that the mid-Pliocene paleogeography still contributes to the increased rainfall over West Africa in mPWP (Stepanek et al., 2020). The enhanced topography with closed Arctic gateways has been shown to strengthen the Atlantic Meridional Overturning Circulation (AMOC), contributing to the warming in the North Atlantic seen in PlioMIP2 as well as the reduced model-proxy mismatch between PlioMIP1 and 2 (Zhang et al., 2020). However, although the strength of the AMOC has been linked to rainfall variability over Sahel (Mulitza et al., 2008), HadGEM3, which did not include the enhanced topography and instead exhibit a weakening of the AMOC (Zhang et al., 2020), still exhibit a rainfall increase over West Africa close to (within 1 standard deviation of) the MMM (Fig. 2). These findings indicate that other mid-Pliocene boundary conditions remain important related to rainfall changes over West Africa.

Land surface changes are also known to impact rainfall over West Africa, where, e.g., expansion of vegetation into the Sahara region at the expense of desert leads to a decrease of the surface albedo and a warming of the region, further strengthening the Sahara Heat Low, and subsequently the WAM, leading to a vegetation-albedo feedback (Charney, 1975). This has been shown to strengthen the response of the WAM to external forcing in other past climates (e.g. Braconnot et al., 1999; Chandan and Peltier, 2020; Claussen and Gayler, 1997; Messori et al., 2019), and the enhanced vegetation in the PlioMIP2 ensemble (Haywood et al., 2020; Salzmann et al., 2008) is likely to have contributed to the strengthening of the mPWP WAM and West African summer rainfall. While vegetation and orography are changed together in PlioMIP2, making it difficult to separate the impact of the two boundary conditions, sensitivity experiments indicate that they play a large role in the mPWP rainfall increase over West Africa (Chan and Abe-Ouchi, 2020; Hunter et al., 2019; Kamae et al., 2016). Studies by Feng et al. (in review, Sci. Adv.) further indicate that the enhanced vegetation drives the mPWP hydroclimatological changes over Sahel to a larger extent than the enhanced topography. In PlioMIP2 the enhanced vegetation is used as a boundary condition (Fig. 8), but a northward expansion of vegetation is also seen in the dynamic vegetation model COSMOS (Stepanek et al., 2020, Fig. 25 e,f), indicating that the mPWP rainfall increase over West Africa and Sahel of 2.5 mm/day for the MMM and 2.3 mm/day for COSMOS (Fig. 3) can support the enhanced vegetation cover seen in the PRISM4.

**Reviewer Comment #2:** Some parts of Africa receive significantly high amounts of precipitation during June and also during October and some of the models simulate a strongly increased precipitation at mid-Pliocene during October. I understand that, with respect to the analysis of the monsoon flow etc., the core summer monsoon season is taken for the calculation of the mean distributions. But, I think the prolongation of the monsoon season is one of the most interesting aspects in the results. Therefore, I recommend to either include both month (or at least October) into the mean, or (if the atmospheric dynamic is significantly different between October and JAS) to present and discuss additionally a plot on the October precipitation and atmospheric circulation.

**Response:** Thank you for this suggestion. Both the precipitation response and the atmospheric dynamics are very similar for JAS and October, and given that these are the four months that exhibit the largest precipitation anomalies in Sahel we have as per your suggestion included October into the seasonal mean (L. 150-153).

**Revised:** All models show an increase in rainfall in the July-October period with the largest increase occurring either in August, September or October, resulting in a lengthening of the WAM. We will therefore base our spatial analysis of the WAM on the July-October (JASO) period, although this does not alter the spatial patterns compared to a shorter monsoon season (July-September).

**Specific comments:**

**Reviewer Comment #3:** L73: model studies or reconstructions?

**Response:** Thank you for pointing this out, the text on L. 79 has been clarified to indicate that the reference refers to model studies.

**Revised:** While previous **model** studies have shown that the high-latitude warming has…

**Reviewer Comment #4:** Fig.1: The different colours are not easy to distinguish, maybe you can think about using different line types for similar colours. In the right panel, the line for the observation cannot be seen, it would help to plot the MMM and modern line on Top of the others.

**Response:** The line styles and colours in Fig. 1 have been altered to make it easier to distinguish between the different models, and the MMM and Modern lines have been shifted to be on top of the lines for the individual models.

**Reviewer Comment #5:** Fig.1: One of the most interesting question is the WAM progression into the Sahara Desert. Please think about including also a seasonal cycle plot for the Sahara (e.g., 20°N-30°N, 20-30°E).

**Response:** Thank you for this suggestion, we have added the seasonal cycle for the Sahara region to Fig. 1.

**Reviewer Comment #6:** Fig.1: In the modern precipitation distribution, the isohyets are tilted, i.e., on the same latitude, rainfall is higher in the western part than in the eastern part of North Africa. Due to this, the region used for analysing the seasonal cycle is often limited to 10°W-10°E.

**Response:** Thank you for pointing this out. Indeed, the isohyets over West Africa are tilted in the modern climate which might motivate a more limited region. However, when comparing the seasonal cycles (pre-Industrial ensemble means shown below) we can see that they are virtually identical for the two cases (10°W-10°E vs. 20°W-30°E). The magnitude of rainfall is slightly higher with the smaller region, but the seasonal distribution is the same. Considering the models have different horizontal resolution we also prefer to use a larger region (L. 122-124).

[Figure]

Fig 1. Seasonal cycle of precipitation in Sahel, Pre-Industrial ensemble means, averaged for a small region (blue) and a bit larger region (orange)

**Revised:** A more narrow definition of the Sahel region is also sometimes used (10°W-10°E, e.g. Thorncroft et al., 2011), but our analysis has shown no difference in the seasonal distribution of rainfall compared to a wider region (20°W-30°E).

**Reviewer Comment #7:** L129: I'm not sure about the quality of the CRU TS v4 data for the early period (1901-1930), because weather stations and the data coverage was and is still very rare in this region. Maybe you can check if there are large differences to the 1960-1990 period, and if you can see interpolation residues.

**Response:** We agree with the reviewer, there is indeed a lack of spatial coverage in precipitation observations over West Africa for the earlier part of the 20th century. However, there are no large

differences in the seasonal cycle between the 1901-1930 and 1961-1990 periods. The later period has slightly lower levels of rainfall in the monsoon season (approx. 0.5 mm/day difference in August), but remains above the PI ensemble mean.

[Figure]

Fig 2. Seasonal cycle of precipitation in Sahel, 1901-1930 mean (blue) and 1961-1990 mean (orange) based on CRU TS v.4 data.

**Reviewer Comment #8:** L143-151: For better comparison, you could mention the observed PI rainfall rate.

**Response:** Thank you for pointing this out, we have included a comparison to the observed PI rainfall on L. 154-158.

> **Revised:** Over the Coast of Guinea, the PI simulations show higher levels of rainfall through most of the Northern Hemisphere's spring, summer and fall, with the ensemble mean showing a maximum of 8.1 mm/day occurring in August (Fig. 1). **This is, both in seasonal distribution and amount, comparable to the PI observations which exhibit maximum rainfall of 7.9 mm/day in September. However, while the observations show a slight bi-modal precipitation distribution, with peaks in June and September, the PI MMM has a wider distribution with a peak in August.**

**Reviewer Comment #9:** Fig.2: The colours of the colour-bar are difficult to distinguish and it is not entirely clear which colour stands for which value. The pattern correlation value is very small. You could save space in the panel plots by omitting the coordinates in the individual plots. The land sea mask can be used for orientation and the coordinates can also be estimated from the MMM plot. This would make it possible to enlarge the plots without increasing the size of the overall plot.

**Response:** Thank you, we have enlarged the individual plots and make the information contained more easily distinguishable as per your suggestions.

**Reviewer Comment #10:** Fig.2: I think it would be interesting to discuss and analyse in more detail, why some models produce a very strong increase in rainfall and some do not. Is there a relationship to the prescribed boundary conditions or specific model physics? Is it just the sensitivity?

**Response:** This is indeed an interesting question. However, there is still a lot of uncertainty as to why the response differs within the model ensemble. Most models implement the same boundary conditions (although with slight differences due to, e.g., model resolution), and the models that have not used all the prescribed PlioMIP2 boundary conditions (HadGEM3 and COSMOS) still exhibit a precipitation response over West Africa within 1 standard deviation of the ensemble mean. Haywood et al. (2020) instead hypothesizes that the sensitivity to the Pliocene boundary conditions is mostly related to model parameterisation and initial conditions, and that within model families later model versions (with the same boundary and initial conditions) tend to be more sensitive than earlier version. The precipitation response over West Africa of the models is in many ways similar to their global response, e.g., GISS-E2-1-G exhibits the lowest precipitation sensitivity to the forcing both globally and over West Africa. We have expanded the discussion on this further in the revised version of the manuscript (L. 321-330).

> **Revised: Haywood et al. (2020) instead suggest that the sensitivity of the individual ensemble members to the mid-Pliocene boundary conditions is mostly related to the model parameterization and initial conditions, with model improvements between the two phases also playing a role. Within model families, later model versions, run with the same boundary and initial conditions, tend to be more sensitive than earlier versions (Haywood et al., 2020). This is consistent with our results, where e.g., CCSM4-NCAR exhibits larger rainfall anomalies than CESM1.2 and IPSLCM6A exhibits larger rainfall anomalies than IPSLCM5A and IPSLCM5A2 (Fig. 3).**
>
> **The precipitation response over West Africa of the models is also in many ways similar to their global response, where e.g.,** the weakest rainfall increase in Sahel is seen in GISS-E2-1-G (Fig 2), consistent with the model's low global rainfall response to the mid-Holocene boundary conditions (Haywood et al., 2020).

**Reviewer Comment #11:** L159: WAM instead of WAS?

**Response:** Corrected.

**Reviewer Comment #12:** Fig.3: Please define the region "Sahel" again in the caption of the plot so that you can read the plot without reading the other figure captions or the text.

**Response:** Thank you for the suggestion, the region has been defined in the caption.

**Reviewer Comment #13:** L226: the word 'anomalies' is doubled.

**Response:** Corrected.

**Reviewer Comment #14:** Sec. 4.2: I think the pattern correlation for the modern precipitation distribution is not the best way to prove and summarize model performances, because North Africa has a very zonal and uniform precipitation pattern. Are there any climatic reconstructions that could be used to estimate whether the precipitation distribution calculated for MMM is correct and the increase in precipitation is of the right order of magnitude?

**Response:** Unfortunately, there are no direct precipitation reconstructions over West Africa for the mPWP. The available proxies are more to be used as qualitative or semi-quantitative indicators of mPWP hydroclimate (i.e., wetter, drier), and only a few of these records exist on or around North Africa. This was looked into recently by Ran Feng et al. (manuscript submitted to Sci. Adv.) and we have linked more closely to their results in the revised version (L. 304-311). However, comparing the vegetation pattern in the COSMOS model (which includes dynamic vegetation and whose

precipitation response is close to the ensemble mean) to the reconstructed vegetation cover in PRISM4 (used as boundary conditions in PlioMIP2) can give an indication on if the modelled climate is in agreement with the reconstructed climate, and we have expanded on this in the revised version of the manuscript (L. 394-397).

> **Revised:** The changes are statistically robust and consistent with previous studies on both PlioMIP1 and 2 where the tropics, particularly the Northern Hemisphere monsoon region, is identified as a region with a robust rainfall signal during the mid-Pliocene (Haywood et al., 2020; Li et al., 2018; Pontes et al., 2020; Zhang et al., 2016). **A model/proxy comparison by Feng et al. (in review, Sci. Adv.) has also shown that the wetter conditions seen over Sahel and West Africa in PlioMIP2 (Fig. 2) are consistent with the available qualitative indicators of mPWP hydroclimate, although the magnitude of change cannot be obtained from these proxy datasets and therefore not compared quantitatively to the PlioMIP2 results.**

> **Revised: In PlioMIP2 the enhanced vegetation is used as a boundary condition (Fig. 8), but a northward expansion of vegetation is also seen in the dynamic vegetation model COSMOS (Stepanek et al., 2020, Fig. 25 e,f), indicating that the mPWP rainfall increase over West Africa and Sahel of 2.5 mm/day for the MMM and 2.3 mm/day for COSMOS (Fig. 3) can support the enhanced vegetation cover seen in the PRISM4.**

**Reviewer Comment #15:** L.327: "...our results support a future strengthening of the WAM and rainfall increase over West Africa and Sahel in a high $CO_2$ scenario." I think that it is not necessarily possible to conclude from the analyses for the mid-Pliocene warm period how the WAM will change in the future. The future will experience much higher $CO_2$ levels and it is not guaranteed that this will lead to a permanent expansion of vegetation. This depends on very different factors.

**Response:** Thank you for this comment. We agree with the reviewer that many different factors might impact the future of the precipitation in West Africa, especially the future changes to the vegetation, and we have revised the manuscript to reflect this both in the discussion (L. 404-407) and in the conclusion (L. 418-425).

> **Revised:** … our results support a future strengthening of the WAM and rainfall increase over West Africa and Sahel in a high $CO_2$ scenario. **However, given the role of the enhanced vegetation in the strengthening of the mPWP WAM, this will also depend on the future vegetation changes in the region which still remain elusive (Bathiany et al., 2014).**

> **Revised:** Given the potential for using the PlioMIP2 as an analogue for near-future scenarios, these results suggest a more uniform rainfall increase over West Africa and the Sahel region, unlike the east-west contrast seen in both CMIP3 and CMIP5 future projections (Roehrig et al., 2013). **Alternatively, these results suggest that the extent of analogism between mid-Pliocene and future climate in the context of rainfall over West Africa may depend on the long-term response of vegetation to the $CO_2$ forcing and on the speed with which climate adapts to future carbon dioxide burden - CMIP simulations of a transient climate and mid-Pliocene simulations of a quasi-equilibrium climate representing endmembers of potential future conditions.**

**Author response to Reviewer #2**

**Reviewer Comment #1:** The authors conclude that the strengthened mid-Pliocene WAM is "most likely due to the greenhouse gas forcing". I am not entirely convinced by this main conclusion. Given the large spread in projected future WAM changes in the CMIP3 and CMIP5 ensembles, GHG forcing does not seem very likely as the (only) major cause for the very consistent and robust rainfall changes in the PlioMIP2 ensemble. Instead, I presume that other factors also play a crucial role. In particular, I suspect that the prescribed Pliocene vegetation cover over North Africa plays a key role, which would probably imply circular reasoning when the authors state that Pliocene greening of North Africa indicates wetter conditions, "which is qualitatively consistent with the results from the PlioMIP2 ensemble". I think the authors should provide some stronger arguments to conclude that GHG forcing is the major driver for the stronger mid-Pliocene WAM. I also wonder about the roles of other Pliocene boundary conditions that were applied in these simulations, like lake fraction, soils, a reduced Greenland ice sheet and the land-sea mask (Haywood et al., 2016). Unless sensitivity studies with individual forcings (i.e. boundary condition changes) can be presented, I suggest to perform some more detailed analyzes. For instance, how much do surface albedoes change (see Charney feedback through vegetation-induced albedo changes)? How large is the contribution of local water recycling (e.g. Brubaker et al., 1993)? What about changes in the large-scale meridional temperature gradient, which could be affected by reduced ice sheets and a stronger AMOC, which in turn could be induced by the closing of Bering Strait? A strong AMOC and a warm North Atlantic are well known key drivers of a stronger WAM (e.g. Mulitza et al., 2008). Maybe a combination of different forcing factors can explain the robust wettening of Pliocene North Africa, but I doubt that it is only the effect of GHG.

**Response:** We thank the reviewer for this comment, and agree that indeed GHGs are not solely responsible for the precipitation changes. Vegetation and land-surface changes are known to impact the West African Monsoon and rainfall through, e.g., vegetation-albedo feedbacks (Charney et al., 1975), and North Atlantic sea surface temperatures have long been linked to precipitation changes in the Sahel and West African region, suggesting that non-$CO_2$ boundary conditions play an important role in the mid-Pliocene precipitation response over West Africa. A recent paper by Zhang et al. (2020) has indeed shown a stronger AMOC within the PlioMIP2 ensemble for all models that include closed Arctic gateways. However, HadGEM3, which did not include this enhanced land/sea mask and instead exhibited a weakening of the AMOC still exhibit a precipitation increase over West Africa close (within 1 std) to the ensemble mean, suggesting that, in this case, other boundary conditions (such as vegetation) might play a larger role in the mid-Pliocene precipitation changes over West Africa. The results from different PlioMIP2 sensitivity experiments for many of the different ensemble members have now been published, and we have expanded the discussion on the role of the different non-CO2 boundary conditions (e.g., vegetation, topography) and the changes they induce (e.g., changes to AMOC) on the enhanced monsoonal rainfall in the revised manuscript (L. 367-397).

> **Revised:** In addition to the CO2 forcing, the majority of the PlioMIP2 ensemble members were forced by the PRISM4 boundary conditions, with reconstructed distributions of topography, bathymetry, land ice cover, vegetation and land/sea mask being amongst the major changes compared to modern geography (Dowsett et al., 2016; Haywood et al., 2016). While the PlioMIP2 set-up, with orography and vegetation being changed together, makes it difficult to distinguish between the impact of the two boundary conditions, the sensitivity simulations by COSMOS indicate that the mid-Pliocene paleogeography still contributes to the increased rainfall over West Africa in mPWP (Stepanek et al., 2020). The enhanced topography with closed Arctic gateways has been shown to strengthen the Atlantic Meridional Overturing Circulation (AMOC), contributing to the warming in the North Atlantic seen in PlioMIP2 as well as the reduced model-proxy mismatch between PlioMIP1 and 2 (Zhang et al., 2020). However, although the strength of the AMOC has been linked to rainfall variability over Sahel (Mulitza et al., 2008), HadGEM3, which did not include the enhanced topography and instead exhibit a weakening of the AMOC (Zhang et al., 2020), still

exhibit a rainfall increase over West Africa close to (within 1 standard deviation of) the MMM (Fig. 2). These findings indicate that other mid-Pliocene boundary conditions remain important related to rainfall changes over West Africa.

Land surface changes are also known to impact rainfall over West Africa, where, e.g., expansion of vegetation into the Sahara region at the expense of desert leads to a decrease of the surface albedo and a warming of the region, further strengthening the Sahara Heat Low, and subsequently the WAM, leading to a vegetation-albedo feedback (Charney, 1975). This has been shown to strengthen the response of the WAM to external forcing in other past climates (e.g. Braconnot et al., 1999; Chandan and Peltier, 2020; Claussen and Gayler, 1997; Messori et al., 2019), and the enhanced vegetation in the PlioMIP2 ensemble (Haywood et al., 2020; Salzmann et al., 2008) is likely to have contributed to the strengthening of the mPWP WAM and West African summer rainfall. While vegetation and orography are changed together in PlioMIP2, making it difficult to separate the impact of the two boundary conditions, sensitivity experiments indicate that they play a large role in the mPWP rainfall increase over West Africa (Chan and Abe-Ouchi, 2020; Hunter et al., 2019; Kamae et al., 2016). Studies by Feng et al. (in review, Sci. Adv.) further indicate that the enhanced vegetation drives the mPWP hydroclimatological changes over Sahel to a larger extent than the enhanced topography. In PlioMIP2 the enhanced vegetation is used as a boundary condition (Fig. 8), but a northward expansion of vegetation is also seen in the dynamic vegetation model COSMOS (Stepanek et al., 2020, Fig. 25 e,f), indicating that the mPWP rainfall increase over West Africa and Sahel of 2.5 mm/day for the MMM and 2.3 mm/day for COSMOS (Fig. 3) can support the enhanced vegetation cover seen in the PRISM4.

**Reviewer Comment #2:** Please discuss whether the rainfall increases are sufficient to maintain the prescribed Pliocene vegetation cover. If the simulated rainfall increase was too small, the authors should tone down their statement that the PlioMIP2 "results are consistent with geological evidence".

**Response:** Thank you for this suggestion. Comparing the vegetation pattern in the COSMOS model (which includes dynamic vegetation and whose precipitation response over West Africa is close to the ensemble mean) to the reconstructed vegetation cover in PRISM4 (used as boundary conditions in PlioMIP2) can give an indication on if the modelled climate is in agreement with the reconstructed climate and if the precipitation changes are sufficient to sustain the prescribed vegetation cover. The vegetation pattern in COSMOS is generally in agreement with the expected vegetation patterns during the mid-Pliocene (Stepanek et al., 2020), and we have discussed this further in the revised version of the manuscript (L. 394-397).

**Revised:** In PlioMIP2 the enhanced vegetation is used as a boundary condition (Fig. 8), but a northward expansion of vegetation is also seen in the dynamic vegetation model COSMOS (Stepanek et al., 2020, Fig. 25 e,f), indicating that the mPWP rainfall increase over West Africa and Sahel of 2.5 mm/day for the MMM and 2.3 mm/day for COSMOS (Fig. 3) can support the enhanced vegetation cover seen in the PRISM4.

**Reviewer Comment #3:** Regarding WAM dynamics the authors only show SLP and 850h Pa wind anomalies. Other key dynamical features of the WAM, like the AEJ and TEJ are not considered at all, but are known to impact West African summer rainfall. At least a latitudinal transect of mean summer zonal wind over Africa, similar to figure 5 in Nicholson (2013), should be presented to provide a wider picture of the changes in the WAM system.

**Response:** We agree with the reviewer that these key dynamical features are important when it comes to understanding changes to the West African Monsoon. However, the aim of this paper is to evaluate changes to the precipitation, and we believe that a detailed dynamical analysis is outside the scope of this manuscript which focuses on the large-scale patterns of the rainfall within a large ensemble. A more dynamically oriented study is surely worth doing, and we plan to explore this in a future paper using a more limited number of models.

**Reviewer Comment #4:** Line 280: What is the main reason for the stronger rainfall changes in PlioMIP2 compared to PlioMIP1? Is it a change in the boundary conditions or perhaps improvements in the climate models? Please discuss.

**Response:** We thank the reviewer for this comment, it is indeed an important question although one that is difficult to answer with the simulations available within PlioMIP2. Some of the larger changes to the boundary conditions in PlioMIP2 compared to PlioMIP1 is the land/sea mask, but the precipitation response over West Africa in HadGEM3, which did not include this boundary condition, still remains close (within 1 std) to the ensemble mean. Generally, models that exhibit a large temperature sensitivity to the PlioMIP2 forcing and boundary conditions also exhibit a large precipitation response, and within model families later model versions tend to be more sensitive than earlier version (Haywood et al., 2020), indicating that the strength of the increased rainfall changes is more related to model sensitivity and the model improvements that have been done between PlioMIP1 and PlioMIP2. We will address this further in the revised manuscript (L. 316-330).

> **Revised:** The updated boundary conditions from PRISM3 to PRISM4 might have contributed to this enhancement, where a sensitivity study by (Samakinwa et al., 2020) using COSMOS has shown that the updated paleogeography played the largest role in the changes to the global large-scale climate between PlioMIP1 and PlioMIP2. However, these changes appear to be more pronounced in high than low latitudes (Samakinwa et al., 2020), and HadGEM3 and MRI-CGCM 2.3, which did not implement the enhanced boundary conditions, still exhibits a precipitation response over West Africa within 1 standard deviation of the MMM (Fig. 2). Haywood et al. (2020) instead suggest that the sensitivity of the individual ensemble members to the mid-Pliocene boundary conditions is mostly related to the model parameterization and initial conditions, with model improvements between the two phases also playing a role. Within model families, later model versions, run with the same boundary and initial conditions, tend to be more sensitive than earlier versions (Haywood et al., 2020). This is consistent with our results, where e.g., CCSM4-NCAR exhibits larger rainfall anomalies than CESM1.2 and IPSLCM6A exhibits larger rainfall anomalies than IPSLCM5A and IPSLCM5A2 (Fig. 3).
>
> The precipitation response over West Africa of the models is also in many ways similar to their global response, where e.g., the weakest rainfall increase in Sahel is seen in GISS-E2-1-G (Fig 2), consistent with the model's low global rainfall response to the mid-Holocene boundary conditions (Haywood et al., 2020).

---

## Author Response (AR2)

**Author response to Reviewer #1**

On the manuscript

**Mid-Pliocene West African Monsoon Rainfall as simulated in the PlioMIP2 ensemble**

by Ellen Berntell et al., submitted to *Climate of the Past* (https://doi.org/10.5194/cp-2021-16).

**We thank the reviewer the time and effort spent reviewing our manuscript a second time. We have corrected the manuscript based on the comments provided, and below is listed our response to the specific questions, with the reviewer's comments in black and our replies and revised text in blue.**
* * *
**Reviewer Comment #1:** The statement "...leads to a decrease of the surface albedo and a warming of the region..." in line 385 is not necessarily correct. Instead of "warming" better write "increase in equivalent potential temperature". Moreover, alternative vegetation-precipitation feedback mechanisms for West Africa have been suggested by Patricola and Cook (2008, JGR, doi:10.1029/2007JD009608) and Rachmayani et al. (2015, Clim. Past, doi:10.5194/cp-11-175-2015). These studies should also be cited.

**Response:** Thank you for these suggestions, we have corrected the text and expanded the discussion to reflect the additional vegetation-precipitation feedback mechanisms. (L. 384-390).

> **Revised:** Land surface changes are also known to impact rainfall over West Africa, where, e.g., expansion of vegetation into the Sahara region at the expense of desert leads to a decrease of the surface albedo and **an increase in equivalent potential temperature,** further strengthening the Sahara Heat Low, and subsequently the WAM, leading to a vegetation-albedo feedback (Charney, 1975). **Additionally, later modelling studies have emphasized the role of soil moisture (Patricola and Cook, 2008) and evapotranspiration (Rachmayani et al., 2015) in the vegetation-precipitation feedback due to their effect on low-level moist static energy, convective instability and surface latent heat flux anomalies.**

**Reviewer Comment #2:** The statement "This is consistent with our results, where e.g., CCSM4-NCAR exhibits larger rainfall anomalies than CESM1.2 ..." in line 326 is misleading since CCSM4 is older than CESM1.2. Please correct.

**Response:** Thank you for pointing this out, the text has been corrected to indicate that it is the **CESM2** model, released in 2019, that exhibits larger rainfall anomalies than CESM1.2, released in 2013 (L. 326-327).

> **Revised:** This is **also** consistent with our results, where e.g., **CESM2** exhibits larger rainfall anomalies than CESM1.2 …